

# CTRP3 as a novel biomarker in the plasma of Saudi children with autism

Manan Alhakbany[1,*], Laila Al-Ayadhi[1,2] and Afaf El-Ansary[2,3,4,*]

[1] Department of Physiology, Faculty of Medicine, King Saud University, Riyadh, Saudi Arabia
[2] Autism Research and Treatment Center, Riyadh, Saudi Arabia
[3] CONEM Saudi Autism Research Group, King Saud University, Riyadh, Saudi Arabia
[4] Central Laboratory, Female Center for Scientific and Medical Studies, King Saud University, Riyadh, Saudi Arabia
* These authors contributed equally to this work.

## ABSTRACT

**Background:** C1q/tumor necrosis factor-related protein-3 (CTRP3) has diverse functions: anti-inflammation, metabolic regulation, and protection against endothelial dysfunction.
**Methods:** The plasma level of CTRP3 in autistic patients ($n$ = 32) was compared to that in controls ($n$ = 37) using ELISA.
**Results:** CTRP3 was higher (24.7% with $P < 0.05$) in autistic patients than in controls. No association was observed between CTRP3 and the severity of the disorder using the Childhood Autism Rating Scale (CARS). A positive correlation between CARs and the age of patients was reported. Receiver operating characteristic (ROC) analysis demonstrated a low area under the curve (AUC) for all patients (0.636). Low AUCs were also found in the case of severe patients (0.659) compared to controls, but both values were statistically significant ($P \leq 0.05$). Despite the small sample size, we are the first to find an association between CTRP3 and autism spectrum disorder (ASD).

## INTRODUCTION

Autism spectrum disorder (ASD) is a biologically neurodevelopmental disorder afflicting about one in every 59 children, and it is expected to increase globally (*Bjørklund et al., 2018*). The increasing prevalence of ASD made it a high priority for scientists and health care providers and attracted public attention (*Sheldrick & Carter, 2018*; *Xu et al., 2018*). Its behavior diagnoses are based on a triad of symptoms, including impairment in communication, impairment in sociability, and abnormal and stereotypical behavior (*Bjørklund et al., 2018*). These core symptoms can be detected before 3 years old and last for the whole lifetime (*Andres, 2002*).

Recent studies mainly focus on the mechanism and the pathogenesis of ASD. Many biomarkers, which are noninvasive quantitative measures, gave precise indications to specific mechanisms that can be used to provide a better understanding of the etiological

Corresponding author
Manan Alhakbany,
malhakbany@ksu.edu.sa

mechanisms of autism and thereafter its treatment. Blood is considered a potential source for detecting many diseases because it contains enormous numbers of proteins associated with the physiology or pathology of diseases (*Fang et al., 2021*). Finding valid and predictive biomarkers for this disorder will improve earlier diagnosis and intervention. Until now, no specific biomarker has been found to cause autism, but comparing autistic patients with peers without ASD can help to better understand of the disease.

Many biomarkers were related to ASD and had a significant role in its pathogenesis; immunological/inflammatory markers are considered one of these biomarkers. Moreover, it was found that some ASD biomarkers were generated from lipid abnormalities (*El-Ansary & Al Dera, 2016*).

C1q/tumor necrosis factor (TNF)-related proteins (CTRPs) family, a paralogue of adiponectin, was discovered. There are 15 members extending from CTRP1 to CTRP15. Each member comprises four different domains, an N-terminal signal domain, a short variable peptide, a collagen-like peptide, and a C-terminal globular like C1q domain (*Ahima et al., 2006*). Both CTRPs and adiponectin are a portion of the C1q/TNF protein, which are higher in molecular weight due to extra C1q domain proteins (*Yi et al., 2012*).

The CTRP family members have multiple physiological effects on metabolism, inflammation, protection against endothelial dysfunction, and angiogenesis.

CTRP3 is a novel member of this family with multiple biological functions (*Peterson, Wei & Wong, 2010*). It is detected in many tissues and organs, including the heart, liver, adipocytes, cartilage, blood vessels, monocytes, fibroblasts, colon, small intestine, pancreas, kidney, and brain (*Schaffler & Buechler, 2012*; *Zhou et al., 2014*).

CTRP3 is considered a strong proangiogenic and neuroprotective adipokine. CTRP3 attenuated secondary brain injury after intracranial hemorrhage (ICH) in rats; it decreased brain edema, preserved the blood-brain barrier (BBB), reduced neurological deficit, and encouraged focal angiogenesis. It applies for its protective role mainly *via* an AMPK/HIF-1α/VEGF-signaling pathway (*Wang et al., 2016*). CTRP3 also exerts its protective effect during ICH by inhibiting oxidative stress *via* PKA/NADPH signaling (*Yang et al., 2017*).

Whether CTRP3, which is a member of the recently discovered adipokine family, acts as a promoter or inhibitor of ASD has not been studied before. Therefore, the goal of this work was to measure the level of CTRP3 in autistic children and compare them with peers without ASD.

## MATERIAL AND METHODS

### Study population

This case-control study was conducted on 32 children diagnosed with ASD according to the 5th edition of the diagnostic and statistical manual of mental disorders criteria (*American Psychiatric Association, 2013*). Patients were recruited from the Autism Research and Treatment Centre, Department of Physiology, King Saud University, Riyadh, Saudi Arabia. The autistic group comprises 32 males ranging between 3 and 12 years (mean ± SD = 7.98 ± 2.59 years). Patients who were associated with neurological disease (such as palsy and tuberous sclerosis), metabolic disorders (*e.g.*, phenylketonuria,

diabetes), and autoimmune disease were excluded from the study, as metabolic disorders and autoimmunity may influence the results of plasma CTRP3 levels.

The control group was formed of 37 age- and sex-matched healthy children; they were collected as previously described by *Mostafa & Al-Ayadhi (2015)*. They were not related to the children with ASD, and their ages ranged between 3 and 12 years (mean ± SD = 7.83 ± 2.64 years). Moreover, they had no clinical findings indicative of immunological, diabetic, chronic diseases, or neuropsychiatric disorders. The control group was the healthy older brothers of the healthy children who visit the Well Baby Clinic, King Khalid University Hospital, Faculty of Medicine, King Saud University, Riyadh, Saudi Arabia for their regular growth assessment (*Mostafa & Al-Ayadhi, 2015*).

This study received approval from the Ethical Committee of King Khalid University Hospital (E-10-220). The parents or the legal guardians signed informed written consent before they were included in the study.

## STUDY MEASUREMENTS

### Clinical evaluation of autistic patients

The clinical evaluation depends on history taken from caregivers, clinical examination, and neuropsychiatric evaluation. Childhood Autism Rating Scale (CARS) was used to assess the severity of the disease (*Mick, 2005*). This scale rates the child from one to four in each of 15 areas (relating to people; emotional response; imitation; body use; object use; listening response; fear or nervousness; verbal communication; non-verbal communication; activity level; level and consistency of intellectual response; adaptation to change; visual response; taste, smell and touch response and general impressions) (*Ozonoff, Boodlin-Jones & Solomon, 2005*). According to the scale, children who have scored 30–36 have mild to moderate ASD ($n = 9$), while those with scores ranging between 37 and 60 points have a severe degree of ASD ($n = 21$) (*Schopler, Reichler & Renner, 1988*).

### Blood sample collection

Ten ml of blood sample was collected in anticoagulant (EDTA) tubes from all the participants after overnight fasting as was previously described by *Qasem, Al-Ayadhi & El-Ansary (2016)*. Plasma and RBCs were collected after centrifugation at 1,000 rpm and stored at specific temperatures till used for analysis.

### Assessment of plasma CTRP3 level

Levels of trimeric CTRP3 (<100 kDa) were measured by BioVendor human CTRP3 ELISA according to the manufacturer's protocol, with an intra-assay coefficient of variation of less than 10%. Samples with CTRP3 levels below the detection limit of the assay were assigned the lowest detectable value (0.001 μg/ml). For accuracy, all samples were investigated as duplicate independent assays to avoid inter-assay variations and to guarantee the reproducibility of the obtained results ($P > 0.05$).

### Statistical analysis

A software program was used for the statistical analysis, and results were expressed as mean ± S.D. All statistical comparisons were made by independent Student's t-test, with

**Table 1 Demographic of children with autism and healthy control.**

|  |  | Children with autismn (*n* = 32) | Control group (*n* = 37) |
|---|---|---|---|
| Age (in years) | Range | 3–12 | 3–12 |
|  | Mean ± SD | 7.98 ± 2.59 | 7.83 ± 2.64 |
| Sex | Male | 32 males | 37 males |
| CARS scores | Mild to moderate (30–36) | 9 | – |
|  | Severe (37–60) | 21 | – |

**Table 2 Test of normality using Shapiro test.**

| Parameters | Groups | N | P value |
|---|---|---|---|
| CTRP3 (ug/ml) | Control | 37 | 0.005 |
|  | Patients | 32 | 0.033 |
| CTRP3 (ug/ml) | Mild to moderate | 8 | 0.041 |
|  | Severe | 20 | 0.153 |
| Age (Years) | Mild to moderate | 5 | 0.037 |
|  | Severe | 15 | 0.269 |

**Note:**
If *P* value less than or equal to 0.05 then the data is not normal distributed and If *P* value more than 0.05 then the data is normal distributed.

*P* value < 0.05 considered significant. A Wilcoxon-Mann Whitney test is usually used when data are not normally distributed (Shapiro-Wilk's test negative). The relationship between the CTRP3, CARs, and age was calculated using the Spearman correlations test, and positive or negative correlations are listed. The receiver operating characteristics (ROC) analysis was performed as an excellent statistical tool for assessing the effectiveness of biomarkers by using the same computer software. The area under the curve (AUC) was calculated to find how a plasmatic marker can discriminate between ASD patients and healthy control participants.

# RESULTS

## Demographic data

The demographic characteristics of children with ASD and their matched controls are shown in Table 1. Among the ASD patients, 9/30 were mild-moderate, and 21/30 were severe.

## Levels of CTRP3 in total and subgroups of ASD patients compared to neurotypical healthy controls

The Shapiro test and boxplot showed that the CTRP3 data in children with ASD and healthy controls were not normally distributed, with *P* values of 0.03 and 0.005, respectively (Table 2). Tables 3 and 4 and Fig. 1 demonstrate the significant increase in CTRP3 in the plasma of children with ASD compared to age- and sex-matched controls. ASD patients recorded 24.7% higher CTRP3 plasma levels than healthy controls (*P* < 0.05).

**Table 3 Mean ± S.D. of CTRP3 in plasma of total autistic patients compared to control subjects.**

| Groups | N | Min. | Max. | Mean ± S.D. | Percent change | *P value |
|---|---|---|---|---|---|---|
| Control (ug/ml) | 37 | 0.12 | 0.68 | 0.33 ± 0.14 | 100.00 | <0.050 |
| Patients (ug/ml) | 32 | 0.13 | 0.99 | 0.41 ± 0.18 | 124.71 | |

Note:
* Comparing between groups using Mann-Whitney test.

**Table 4 Mean ± S.D. of (Ug/ml) in plasma of Mild to Moderate and Severe autistic patients.**

| Groups | N | Min. | Max. | Mean ± S.D. | Percent change | P value |
|---|---|---|---|---|---|---|
| Mild to moderate | 8 | 0.20 | 0.58 | 0.31 ± 0.12 | 100.00 | 0.127 |
| Severe | 20 | 0.13 | 0.99 | 0.43 ± 0.20 | 138.16 | |

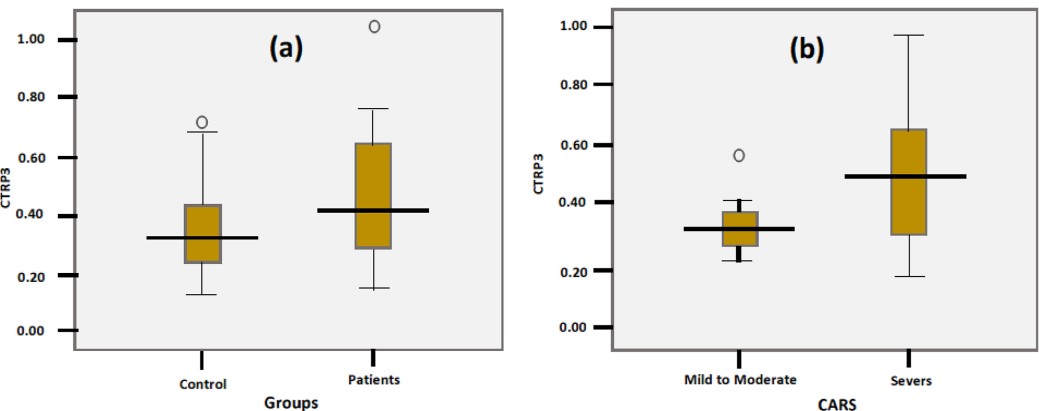

**Figure 1 Boxplot to show the data distribution of CTRP3 in the control and patients' groups (A), and data distribution of CTRP3 in mild to moderate and sever autistic groups (B).**

At the same time, there was no significant difference between mild-moderate and severe autistic children ($P < 0.127$), despite the 38% recorded increase, as shown in Table 4. Boxplots (Fig. 1) show data distribution in control, total ASD, mild-moderate, and severe ASD patients. CTRP3 level, the standard deviation, and the box length as a measure of data dispersion of the ASD patients are considerably higher than control healthy participants and of the severe compared to mild-moderate patients.

## Spearman correlations between CTRP3, age, and CARS

Table 5 and Figure 2 present the Spearman correlations among CTRP3, age, and CARS. Whereas this marker did not show any independent correlations with age and CARS ($P < 0.092$ and $0.750$, respectively), both variables (age and CARS) were negatively correlated ($P < 0.044$). The partial correlation between CTRP3 and the CARS while controlling for age was insignificant, with a correlation co-officiant of $-0.210$ ($P = 0.361$).

**Table 5 Spearman correlations between CTRP3, CARs.**

| Parameters | R (Spearman correlation) | P value | |
|---|---|---|---|
| CTRP3 with Age | −0.368 | 0.092 | N[b] |
| CTRP3 with CARS | 0.061 | 0.750 | P[a] |
| Age with CARS | −0.433* | 0.044 | N[b] |

Notes:
* Correlation is significant at the 0.05 level.
[a] Positive Correlation.
[b] Negative Correlation.

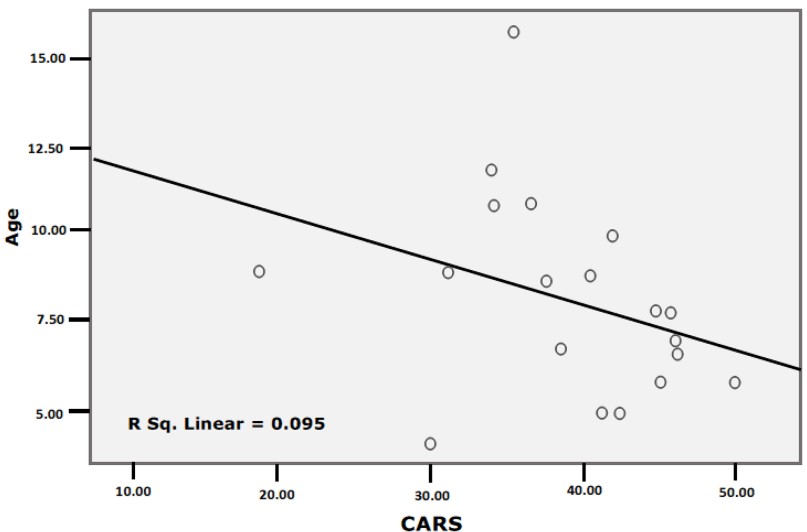

**Figure 2 Correlation between CARS and Age with best fit line curve (negative correlation).**

**Table 6 ROC-Curve of CTRP3 in all patients group according to control group.**

| CTRP3 ROC | AUC | Cut-off value | Sensitivity % | Specificity % | P value | 95% CI |
|---|---|---|---|---|---|---|
| All patients according to control | 0.636 | 0.341 | 56.2% | 70.3% | 0.050 | [0.503–0.769] |
| Mild to moderate according to control | 0.532 | 0.315 | 75.0% | 43.2% | 0.778 | [0.326–0.738] |
| Severe according to control | 0.659 | 0.341 | 65.0% | 70.3% | 0.049 | [0.504–0.814] |
| Severe according to mild to moderate | 0.688 | 0.372 | 60.0% | 87.5% | 0.127 | [0.479–0.896] |

## Data from ROC analysis

Table 6 demonstrates the AUCs, specificity, and sensitivity of all ASD patients and the mild-moderate and severe subgroups compared to control subjects and severe autistic patients compared to mild-to-moderate participants. AUC of 0.9–1.0 shows an excellent predictive value of a biomarker, 0.8–0.9 means a perfect marker, 0.6–0.7 means a good marker, and 0.6 means a useless marker. Based on the AUC ranges, the AUC of all patients compared to controls (AUC of 0.636) and that of severe autistic patients compared to controls (AUC of 0.659) is low but within the acceptable value (AUC of 0.6–0.7).

## DISCUSSION

C1q TNF-related protein 3 (CTRP3) is a relatively novel hormonal factor primarily derived from adipose tissue and plays a role in early childhood development (*Kwon et al., 2018*). In the present study, the remarkably higher level of CTRP3 in plasma of autistic patients compared to healthy and gender matching controls can be easily related to the disruption of BBB as a phenotypic feature in ASD. As a TNF-related protein, the increase of CTRP3 is in good agreement with multiple studies in which TNF-α was significantly increased in the blood and positively correlated with the severity of ASD (*Xie et al., 2017*).

In addition, *Jyonouchi, Sun & Le (2001)* found that TNF-α was elevated in the autistic subjects, and most of those autistic children exhibited excessive or poorly regulated innate immune responses. Moreover, *Chez et al. (2007)* found increased TNF-α in the cerebrospinal fluid of autistic children, and it was also significantly increased in the brains of autistic subjects (*Li et al., 2009*). Although TNF-α was decreased in Saudi autistic patients compared to peers without ASD, this was attributed to the early increase in plasma followed by efflux to the brain through the BBB (*El-Ansary & Al-Ayadhi, 2014*). It is well known that binding with ligand TNF-α can activate NF-κB, MAPK, and the apoptosis signaling pathway (*Perry et al., 2001*). Based on this, the significant increase of CTRP3 in the present study can be easily related to the impaired NF-κB, MAPK, and activated apoptosis signaling pathway repeatedly recorded in autistics compared to peers without ASD (*Young et al., 2011*, *2012*; *Naik et al., 2011*). A study done by *Qasem et al. (2018)* reported a significant increase of NF-κB in plasma of Saudi patients with autism. *Xu, Li & Zhong (2015)* suggest that TNF-α might affect the progress of ASD through another pathway, such as the MAPK/JNK pathway, which can be related to the increase of related CTRP3 reported in the present study. *Gomez-Fernandez et al. (2018)* showed no significant difference in the level of the expression of relevant plasma cytokines, cell adhesion molecules, or growth factors in children with ASD compared with peers without ASD.

CTRP 3 is induced during the late stage of adipocyte differentiation and triggers the secretion of adiponectin with certain regulatory metabolic functions (*Yi et al., 2012*). Current data proves that CTRP3 is functionally the most similar homolog of adiponectin (*Schaffler & Buechler, 2012*). Intravenously injected CTRP3 can cross the BBB and increase the adiponectin levels in the cerebrospinal fluid (*Wang & Scherer, 2016*). Recently, it was found that total CTRP3 concentrations were significantly positively correlated with total cholesterol and HDL cholesterol in children aged 7–10 years (*Alamian et al., 2020*). Based on this finding, the reported increased level of CTRP3 in autistic patients (Tables 3 and 4) supports the association between CTRP3 and metabolic diseases. In addition, it could support the identified subgroups of individuals with ASD characterized by dyslipidemia (*Luo et al., 2020*; *Luçardo Jda et al., 2021*).

Protein kinases are essential in G-protein-coupled, receptor-mediated signal transduction and are involved in neuronal functions, gene expression, memory, and cell differentiation. The cAMP–PKA pathway is one of the most common signaling pathways (*Castro et al., 2013*). The activity and expression of protein kinase A (PKA), a cyclic

AMP-dependent protein kinase, in different areas of the postmortem brain of individuals with ASD demonstrated a significantly lower PKA level than healthy control subjects. The PKA signaling can counteract superoxide anion accumulation and prevent SOD and catalase inhibition that induced by oxidative stress in cultured astrocytes (*Douiri et al., 2016*). Based on the fact that Saudi autistic children are under $H_2O_2$ oxidative stress due to overexpression of SOD with slightly lower catalase, cAMP-PKA can be given special attention (*Al-Gadani et al., 2009*). PKA signaling mediates CTRP 3's anti-oxidative effects during brain injury have yet to be understood. PKA is involved in CTRP 3-mediated suppression of ROS in endothelial cells (*Goldstein, Scalia & Ma, 2009*; *Yang et al., 2017*) confirmed a role for PKA in the protective effects of CTRP 3's against brain injury. It is commonly known that the BBB plays an essential role in protecting the brain by limiting the influx of circulating harmful solutes, macromolecules, and cells from the blood into the brain. However, numerous studies have revealed that dysfunction of the BBB is associated with the pathogenesis of neurological disorders, including ASD, suggesting that some ASD-related proteins might be secreted from the brain into the blood as potential biomarkers (*Theoharides & Doyle, 2008*; *Theoharides et al., 2008*; *Theoharides & Zhang, 2011*; *Fiorentino et al., 2016*).

Based on this, the remarkable increase of plasma CTRP3 in the present study can be related to BBB disruption and brain oxidative stress as confirmed etiological mechanisms in ASD. With a disrupted BBB as an autistic feature, the recorded increase of CTRP3 could be concomitant with a remarkable decrease in the brain due to efflux from the brain to blood. A study by *Rai-Bhogal et al. (2018)* confirmed the inhibition of PKA signaling as a pre-requisite of CTRP3 protective effects with prostaglandins as a proinflammatory lipid mediator. Prostaglandins were among the elevated lipid mediators previously reported by *El-Ansary & Al-Ayadhi (2012)* in plasma of Saudi autistic patients compared to peers without ASD. In a most recent study done by *Qasem et al. (2018)*, PGE2 and mPGES-1 were positively correlated with NF-κB as a proinflammatory marker and associated with the dysfunction in sensory processing.

Table 6 demonstrates the AUCs, specificity, and sensitivity of all ASD patients compared to control subjects and severe autistic patients compared to mild-to-moderate participants. The recorded AUC of all patients compared to controls (AUC of 0.636) and that of severe autistic patients compared to controls (AUC of 0.659) are low but within the satisfactory value known for ROC AUC (AUC of 0.6–0.7), help to accept CTRP3 as a novel predictive biomarker of ASD.

## CONCLUSION

CTRP3 is a novel biomarker that had never been measured in plasma of patients with ASD; CTRP3 was higher in autistic patients than controls. Therefore, it has a role in the early diagnosis of this disorder.

The current study's data are hindered by the relatively small sample number of participants; therefore, it needs to be replicated by a larger size of both individuals with ASD and controls. Also, all the recruited subjects in this study were males, so the

conclusions cannot be anticipated in females with ASD. The inclusion of both sexes in our future studies may help to clarify the sex differences in this disorder.

## ACKNOWLEDGEMENTS

The authors want to extend their thanks to the Autism Research and Treatment Center, Riyadh, Saudi Arabia, for providing blood samples from children with autism and the control group. Special thanks to all the participants and their families for taking part in this study.

### Funding

This project was funded by the National Plan for Science Technology and Innovation (MAARIFAH), King Abdulaziz City for Science and Technology, Saudi Arabia, Award number: 08-MED 510–02. The funders had no role in study design, data collection and analysis, decision to publish, or preparation of the manuscript.

### Grant Disclosures

The following grant information was disclosed by the authors:
National Plan for Science Technology and Innovation (MAARIFAH).
King Abdulaziz City for Science and Technology, Saudi Arabia: 08-MED 510–02.

### Competing Interests

The authors declare that they have no competing interests.

### Author Contributions

- Manan Alhakbany conceived and designed the experiments, performed the experiments, analyzed the data, prepared figures and/or tables, authored or reviewed drafts of the paper, and approved the final draft.
- Laila Al-Ayadhi conceived and designed the experiments, authored or reviewed drafts of the paper, and approved the final draft.
- Afaf El-Ansary conceived and designed the experiments, analyzed the data, prepared figures and/or tables, authored or reviewed drafts of the paper, and approved the final draft.

### Human Ethics

The following information was supplied relating to ethical approvals (*i.e.*, approving body and any reference numbers):

This study received approval from the Ethical Committee of King Khalid University Hospital (E-10-220).

### Data Availability

The ElISA measurements for both children with autism and the control group are available in the Supplemental File.

## Supplemental Information

Supplemental information for this article can be found online at http://dx.doi.org/10.7717/peerj.12630#supplemental-information.

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
