# Peer review of "CTRP3 as a novel biomarker in the plasma of Saudi children with autism"

_PeerJ, doi:10.7717/peerj.12630_

## Round 0.1 · original submission · Major Revisions

Please address critiques of both reviewers and amend your manuscript accordingly.

Please note that although one of the reviewers requested to cite some latest studies or reviews of ASD diagnostic markers in the Introduction section, you do not need to include the specified citations.

Reviewer 1 ·

Basic reporting

The English language should be improved to ensure that an international audience can clearly understand your text. There were multiple grammatical errors and font sizes. Furthermore, several sentences did not transition well.

More information should have been provided, i.e. expanding on the identification of biomarkers.

Experimental design

Which type of CRTP3 was measured in this study? Please specify
CTRP3 can circulate in different oligomeric complexes: trimeric (<100 kDa), middle molecular weight (100–300 kDa), and high molecular weight (HMW) oligomeric complexes (>300 kDa).

The design of the experiments did not have depth to them. More experiments could have been done to further expand or support on their very minimal results section.

Validity of the findings

-The results section is only one paragraph? Results section should be further developed. Please provide more information to help the readers understand the results, the 6 tables and figure 1. A well-constructed Results section includes subsections that describe a question, the experiment conducted to answer that question, the results of that experiment, and the answer to the question. Subheadings provide visual signals at the start of descriptions of separate experiments to help guide readers.

Additional comments

Overall this paper has a lot more work that should be done before it can be considered for publication. Annotations have been made on the pdf.

Annotated reviews are not available for download in order to protect the identity of reviewers who chose to remain anonymous.

Reviewer 2 ·

Basic reporting

no comment

Experimental design

no comment

Validity of the findings

no comment

Additional comments

The paper entitled “CTRP3 as novel biomarker in plasma of Saudi children with autism” aims to explore whether CTRP3 could be used as a novel biomarker for ASD. This topic is interesting. I would recommend the authors to address the following points when revising the manuscript.
Major points
[1] CTRP3 can be detected in a variety of tissues and organs (lines 64 to 67), however, in lines 138-140, the author states that “the remarkable higher level of CTRP3 in plasma of autistic patients compared to healthy and gender matching controls can be easily related to the disruption of BBB as phenotypic feature in autism"; in lines 173-174, “With a disrupted BBB as autistic feature, the recorded increase of CTRP3 could be concomitant with remarkable decrease in the brain due to influx from brain to blood.” How to explain this?
[2] If there are some new literatures, the literature related to CTRP3 need to be updated. The references in this manuscript are from 2018 and before. In addition, are there any studies related to this study in recent years?
[3] The patients in this manuscript were diagnosed through the 5th edition of the diagnostic and statistical manual of mental disorders criteria. Therefore, it is better to use ASD instead of autism.
[4] In the results section, the author did not tell the unit of the CTRP3 measurement value.
[5] I recommend the author to cite some latest studies or reviews of ASD diagnostic markers in the Introduction section, such as Shen et al., 2019, DOI: https://doi.org/10.1007/978-3-030-05542-4_11; Shen et al., 2020, DOI: https://doi.org/10.1016/j.cca.2019.12.009
Minor points
[1] It is very strange, in lines 64 to 71, there are some yellow highlights. Please correct.
[2] Figure 1 is not clear enough, and the display of significant differences in this figure is also unclear. Figure 2 is also not clear enough.
[3] The "P" of the P value, whether in uppercase or lowercase, please unify the full text.
[4] In line 167, it should be “H2O2”.
[5] Please revise carefully to eliminate grammatical errors.

---

## Round 0.2 · Minor Revisions

Please address remaining concerns of the reviewers and revise manuscript accordingly.

Reviewer 1 ·

Basic reporting

I thank the authors for addressing the previous comments.

However, please consider revising the use of your English language. Despite the attached certification provided to ensure that your English has been corrected, there are multiple areas that need to be worked on. Please note that there are several grammatical mistakes and poor sentence structures.

Experimental design

It is great that the authors are looking for biomarkers and despite a small cohort size, have been able to identify a potential marker. They covered the design well.

Validity of the findings

All data has been described well.

Additional comments

None

Reviewer 2 ·

Basic reporting

no comment

Experimental design

no comment

Validity of the findings

no comment

Additional comments

The article has improved. However, some grammar still needs to be revised. Please check carefully. In addition, the author mentioned that the revisions were highlighted in yellow, which I did not see. Moreover, the point-to-point answers have not been fully completed. For example, the 2 in "H2O2" should be a subscript.

---

## Round 0.3 · accepted · Accept

All remaining issues were addressed and the manuscript was amended accordingly. Therefore, revised version is acceptable now.